# Recombinant Human Annexin A5 Alleviated Traumatic-Brain-Injury Induced Intestinal Injury by Regulating the Nrf2/HO-1/HMGB1 Pathway

**DOI:** 10.3390/molecules27185755

**Published:** 2022-09-06

**Authors:** Hejun Zhang, Yalong Gao, Tuo Li, Fanjian Li, Ruilong Peng, Cong Wang, Shu Zhang, Jianning Zhang

**Affiliations:** 1Department of Neurosurgery, Tianjin Medical University General Hospital, Tianjin 300000, China; 2Graduate School, Tianjin Medical University, Tianjin 300000, China; 3Key Laboratory of Post-trauma Neuro-repair and Regeneration in Central Nervous System, Ministry of Education, Tianjin 300000, China; 4Department of Neurosurgery, First Hospital of Qinhuangdao, Qinhuangdao 066000, China; 5Department of Neurosurgery, Yantai Yuhuangding Hospital, Yantai 264000, China

**Keywords:** ANXA5, HMGB1/Nrf2/HO-1, inflammatory, intestine, oxidative stress

## Abstract

**Aims:** Annexin A5 (ANXA5) exhibited potent antithrombotic, antiapoptotic, and anti-inflammatory properties in a previous study. The role of ANXA5 in traumatic brain injury (TBI)-induced intestinal injury is not fully known. **Main methods:** Recombinant human ANXA5 (50 µg/kg) or vehicle (PBS) was administered to mice via the tail vein 30 min after TBI. Mouse intestine tissue was gathered for hematoxylin and eosin staining 0.5 d, 1 d, 2 d, and 7 d after modeling. Intestinal Western blotting, immunofluorescence, TdT-mediated dUTP nick-end labeling staining, and enzyme-linked immunosorbent assays were performed 2 days after TBI. A series of kits were used to assess lipid peroxide indicators such as malonaldehyde, superoxide dismutase activity, and catalase activity. **Key findings:** ANXA5 treatment improved the TBI-induced intestinal mucosa injury at different timepoints and significantly increased the body weight. It significantly reduced apoptosis and matrix metalloproteinase-9 and inhibited the degradation of tight-junction-associated protein in the small intestine. ANXA5 treatment improved intestinal inflammation by regulating inflammation-associated factors. It also mitigated the lipid peroxidation products 4-HNE, 8-OHDG, and malonaldehyde, and enhanced the activity of the antioxidant enzymes, superoxide dismutase and catalase. Lastly, ANXA5 significantly enhanced nuclear factor E2-related factor 2 (Nrf2) and hemeoxygenase-1, and decreased high mobility group box 1 (HMGB1). **Significance:** Collectively, the results suggest that ANXA5 inhibits TBI-induced intestinal injury by restraining oxidative stress and inflammatory responses. The mechanisms involved sparking the Nrf2/hemeoxygenase-1-induced antioxidant system and suppressing the HMGB1 pathway. ANXA5 may be an attractive therapeutic candidate for protecting against TBI-induced intestinal injury.

## 1. Introduction

Traumatic brain injury (TBI) threatens human health worldwide, owing to its extremely high associated disability and mortality rates [1]. Rather than being an isolated event, it is a complicated biological process that is associated with a great deal of extracranial injuries, such as systemic inflammation and multiorgan dysfunction. Cardiovascular, respiratory, and gastrointestinal systems are most frequently affected after TBI [2]. These secondary injuries to other organs are associated with an increased mortality after TBI. There is therefore a need to understand the underlying mechanisms of post-TBI extracranial complications.

Organ dysfunction, particularly gastrointestinal dysfunction, has frequently been reported in previous studies [3]. Gut microbiota and gastrointestinal dysfunction can be induced by TBI [3]. Gastrointestinal dysfunctions mainly include dysmotility, barrier disruption, and mucosal injury along the gastrointestinal tract. In addition, post-traumatic morbidity and mortality could be affected by gastrointestinal dysfunctions that possess the ability of altering immune responses [4,5]. Recent studies demonstrate that TBI can induce intestinal oxidative stress-associated damage [6]. TBI can induce the ischemia or hypoperfusion of the intestinal mucosa, and due, to scant stored energy, intestinal mucosal epithelial cells are very sensitive to ischemia and hypoxia. Excessive free radicals, increased inflammatory factors, apoptosis, and intestinal inflammation infiltration are generated after TBI, leading to the destruction of intestinal mucosal barrier function [7]. Consequently, the insulted intestinal mucosa can cause the translocation of endotoxins and bacteria in the intestinal tract, which can further exacerbate the systemic inflammatory response and even result in multiple organ failure and death [8].

Intestinal damage after TBI is often accompanied by oxidative stress. Nuclear factor E2-related factor 2 (Nrf2) is one of the regulators of cytoprotection against oxidative stress [9]. There are numerous reports that Nrf2 can regulate the innate immune response in various experimental models of disease [10]. Among the enzymes regulated by Nrf2, HO-1 is acknowledged to be capable of anti-inflammatory and anti-oxidative properties [11,12,13]. Upon stimulation by oxidative stress and electrophiles, Nrf2 translocates to the nucleus, binding to a small Maf protein (staff) and immediately forming a heterodimer. This heterodimer subsequently induces the transcriptional activation of numerous antioxidative genes, including catalase and superoxide dismutase (SOD) [14]. Previous studies indicate that improving intestinal injury by regulating the Nrf2/HO-1 pathway is of great significance [15]. Reducing intestinal mucosal damage and maintaining gut function benefit the prognosis of TBI, but few specific drugs target these pathological lesions and reduce intestinal injuries after TBI.

Annexin A5, also known as ANXA5, is a Ca^2+^-dependent phosphatidylserine-binding protein that is widely used to detect phospholipid serine in apoptotic cells. Through the combination of calcium and phospholipid, ANXA5 plays the role of endogenous regulation of a variety of physiological processes. [16]. In previous studies, ANXA5 has exhibited potential anti-inflammatory, antiapoptotic, and antithrombotic effects [17,18,19,20]. It also plays a vital role in the cellular inflammatory response, as it can inhibit local vascular and systemic inflammation [21,22]. However, whether ANXA5 can inhibit TBI-induced intestinal injury has not been thoroughly investigated. The current study investigated protective mechanisms involving ANXA5 in responses to TBI-induced intestinal injury in mice.

## 2. Results

### 2.1. ANXA5 Reduced Intestinal Lesions Induced by TBI

HE staining indicated different degrees of intestinal injury 0.5, 1, 2, and 7 days after TBI (Figure 1A). The damage was most evident on day 2, at which timepoint, Chiu’s scores were significantly lower in the TBI+ANXA5 group than in the TBI group (*n* = 6 mice/group, *p* < 0.05) (Figure 1B).

### 2.2. ANXA5 Increased Mouse Body Weight after TBI

TBI induced all mice to lose weight in the first week post-injury, but the weight in the ANXA5 group was regained more quickly than that in the TBI group (day 7; *p* < 0.05, *n* = 10 mice/group) (Figure 1C).

### 2.3. ANXA5 Inhibited TBI-Induced Intestinal Apoptosis

Apoptotic cells in the lesioned intestine were identified with the help of TUNEL staining. The apoptotic fraction was greater in the TBI group than in the sham group on day 2 after TBI (*p* < 0.05), and the apoptotic proportion was significantly lower in the TBI+ANXA5 group than in the TBI group (*n* = 7 mice/group, *p* < 0.05) (Figure 1D,E). Western blotting revealed that TBI resulted in a significant upregulation of cleaved caspase-3 and Bax in the intestine (*p* < 0.05), and cleaved caspase-3 and Bax levels were significantly lower in the TBI+ANXA5 group than in the TBI group. On the contrary, TBI resulted in a significant reduction in Bcl2; after ANXA5 intervention, the expression of Bcl2 evidently increased (*n* = 6 mice/group, *p* < 0.05) (Figure 1F–I).

### 2.4. ANXA5 Alleviated TBI-Induced Intestinal Mucosal Damage

TBI is linked to altered endothelial function profiles in the intestine. In order to investigate the role of tight junction protein responses after TBI, changes in the endothelial function in the intestine were assessed on day 2 after TBI. Western blotting revealed that occludin (Figure 2A,B) and claudin-1 (Figure 2C,D) were damaged in the TBI group compared with the sham group (*p* < 0.05). Immunofluorescence analysis also revealed that expression levels of occludin (Figure 2E,F) and claudin-1 (Figure 2G,H) were markedly diminished in the TBI group compared with the sham group (*p* < 0.05). The administration of ANXA5 significantly increased the expression levels of occludin and claudin-1 after TBI (*p* < 0.05).

### 2.5. ANXA5 Mitigated Intestinal MMP-9 after TBI

On the 2nd day after TBI, the expression of MMP-9 in the TBI group was significantly higher than that in sham group (*p* < 0.05). ANXA5 administration significantly reduced MMP-9 expression compared with the vehicle-alone group (*p* < 0.05) (Figure 2I,J).

### 2.6. ANXA5 Increased Anti-Inflammatory Cytokines and Reduced Pro-Inflammatory Cytokines after TBI

TBI elevated pro-inflammatory factors in the intestine. In order to investigate the role of inflammatory responses after TBI, the levels of IL-1β, IL-6, TNF-α, and IL-10 were assessed. ELISAs indicated that TBI induced the secretion of the pro-inflammatory cytokines IL-1β and IL-6, and inhibited the anti-inflammatory cytokine IL-10 compared with the sham group (*p* < 0.05) (Figure 3A–C). In contrast, ANXA5 significantly reduced IL-1β and IL-6, and increased IL-10 compared with the TBI group (all *p* < 0.05, *n* = 6 mice per group). There were no significant differences in TNF-α in the three groups.

### 2.7. ANXA5 Inhibited Inflammatory Infiltration after TBI

TBI caused the infiltration of inflammatory cells and activation of inflammatory pathways in the intestine. In order to investigate the role of inflammatory cells and inflammatory markers after TBI, MPO and COX-2-positive cells were assessed in the intestinal parenchyma. Immunofluorescence analysis revealed that MPO and COX-2-labeled cells in ileum tissues were significantly enhanced after TBI compared with the sham group (*p* < 0.05) (Figure 3E–H). ANXA5 treatment was associated with fewer MPO and COX-2-labeled cells in ileum tissues than in the TBI group (all *p* < 0.05, *n* = 6 mice per group). Western blotting indicated that COX-2 expression was markedly elevated after TBI (Figure 3I,J) (*p* < 0.05). Compared with the TBI group, ANXA 5 significantly downregulated the expression of COX-2. (*p* < 0.05, *n* = 6 mice/group).

### 2.8. ANXA5 Attenuated Intestinal Oxidative Stress after TBI

4-HNE is a virulent lipid peroxide end product that is often used as an indicator of it. 8-OHDG, one of the markers of oxidative damage to RNA and DNA, was used to assess the degree of oxidative stress. Changes were assessed to determine whether ANXA5 reduced oxidative damage in the intestine. Immunofluorescence analysis revealed that 4-HNE and 8-OHDG-positive cells were dramatically increased in intestinal mucosa after TBI (Figure 4A–D) (*p* < 0.05, *n* = 6 mice/group). ANXA5 treatment significantly reduced 4-HNE and 8-OHDG-positive cells compared with the TBI group (*p* < 0.05, *n* = 6 mice/group). The effect of ANXA5 on oxidative stress was also assessed by examining the MDA content and the activity of SOD and catalase in the intestine 0.5 and 2.0 days after TBI. TBI significantly increased the MDA content (Figure 4F) (*p* < 0.05) and suppressed SOD and catalase activity (Figure 4E–G) (*p* < 0.05), whereas ANXA5 significantly reduced MDA levels and upregulated SOD and catalase activity compared with the TBI group (all *p* < 0.05).

### 2.9. ANXA5 Reduced HMGB1 and Increased HO-1 and Nrf2 after TBI

HMGB1 is a critical pathogenic factor, and TBI induced a significantly higher HMGB1 expression in the intestine (*p* < 0.05). ANXA5 significantly reduced HMGB1 expression in the intestine after TBI (*p* < 0.05). Western blotting was performed to assess whether ANXA5 protects against intestinal damage via the Nrf2/HO-1 pathway, and ANXA5 upregulated Nrf2 and HO-1 levels compared to the TBI group (*p* < 0.05) (Figure 4H–K).

## 3. Materials and Methods

### 3.1. Mice

C57BL/6 mice were purchased from HFK Bioscience Corporation (Beijing, China). Mice were housed in a specific pathogen-free environment, and all mouse experiments were approved by the Tianjin Medical University Experimental Animal Ethics Committee. Mice were divided into three groups at random; a sham group, a TBI group, and a TBI+ANXA5 group (50 µg/kg).

### 3.2. Fluid Percussion Injury Model

Mice were anesthetized with isoflurane. The fluid percussion injury (FPI) model was generated as previously described. Briefly, a 3 mm diameter cranial cavity was drilled with the dura matter left intact, 2 mm posterior from the bregma and 2 mm lateral to the sagittal suture. A plastic injury cap was sealed around the craniotomy using dental cement, and connected to the FPI device. The pendulum angle of the FPI device was adjusted to achieve a peak pressure of approximately 1.9 ± 0.2 atmospheres when triggered against the capped intravenous tubing. After the experiment, the mouse was immediately removed from the apparatus and the wound was sutured closed. The sham group underwent the same process except for the release of the pendulum. Thirty minutes later, the sham group and the TBI group were administered saline, and the treatment group was administered ANXA5 via the tail vein.

### 3.3. Hematoxylin and Eosin Staining

Ileal segments were collected and fixed. Specimens were embedded in paraffin and cut into 4 μm thick sections. The sections were stained with hematoxylin and eosin (HE) and observed under a light microscope (Olympus BX61, Tokyo, Japan). As reported in the previous literature, Chiu’s scoring system [23] was used to quantify the degree of intestinal damage after TBI.

### 3.4. Lipid Peroxidation Assay

A lipid peroxidation assay kit (S0131S, Beyotime, Beijing, China) was used to assess malondialdehyde (MDA) levels in intestinal lysates, according to the manufacturer’s instructions. Briefly, MDA reacts with chromogenic reagents at 100 °C for 15 min and produces a stable chromophore with a maximum absorption peak at 532 nm. A Catalase Assay Kit (S0051, Beyotime, Beijing, China) was used to assess catalase activity in intestinal homogenates, in accordance with the manufacturer’s instructions. Measurement of SOD activity in tissue homogenates was performed in accordance with the technical manual of the detection kits (Nanjing Jiancheng Bioengineering Institute, Nanjing, China).

### 3.5. Immunofluorescence Staining

Frozen 6 μm thick tissue sections were prepared as described above [24]. Tissue sections were blocked by five percent bovine serum albumin in phosphate-buffered saline (PBS) with 0.1% Triton X-100, which were then incubated at 4 °C overnight with primary antibodies against matrix metalloproteinase-9 (MMP-9) (1:500, rabbit IgG; CST), MPO (1:500, rabbit IgG; CST, Danvers, MA, USA), occludin (1:200, rabbit IgG; Abclonal, Woburn, MA, USA), claudin-1 (1:200, rabbit IgG; Abclonal), COX-2 (1:500, rabbit IgG; Abcam, Cambridge, UK), 4-HNE (1:200, rabbit IgG; Abcam), and 8-OHDG (1:200, rabbit IgG; Santa Cruz, Dallas, TX, USA). After being rinsed four times with PBS for 5 min, the sections were then incubated with secondary antibody (1:500, Dylight488 goat anti-rabbit IgG, Abcam) for 1 h. After being washed three times with PBS, a mounting medium containing 4′,6-diamidino-2-phenylindole (DAPI) was added to the sections, which were then sealed with a coverslip. Fluorescent signals were observed under a fluorescence microscope (Olympus BX61, Tokyo, Japan). Three areas were selected from each slide for analysis, and mean values were calculated.

### 3.6. Enzyme-Linked Immunosorbent Assay

Intestinal samples were homogenized in accordance with the manufacturer’s protocol. Enzyme-linked immunosorbent assay (ELISA) kits were used for detecting inflammatory factors in the samples, including tumor necrosis factor (TNF) α and interleukin (IL) 1β, IL-6, and IL-10 (all from R&D Systems, Minneapolis, MN, USA). Measured OD values were converted into a concentration value.

### 3.7. TUNEL Staining

An in situ cell death detection kit (Beyotime, Beijing, China) was used to assess apoptosis in the intestine in accordance with the manufacturer’s instructions. Briefly, after being incubated in 0.125% trypsin for 20 min at 37 °C, the slides were permeabilized with 0.5% Triton X-100 for 30 min. After incubation with TdT enzyme solutions for 1 h at 37 °C, the slides were counterstained with DAPI for 15 min, mounted, and viewed using a fluorescence microscope (Olympus BX61, Tokyo, Japan).

### 3.8. Western Blotting

Western blotting was performed as previously described [25]. Lysis buffer with added protease and phosphatase inhibitors were used for extracting total protein from intestinal tissues in each group (Beijing Solarbio Science and Technology Co., Ltd., Beijing, China). After grinding tissue, the supernatant protein was collected and quantified via the bicinchoninic acid method. PVDF membranes were incubated with specific primary antibodies, which included rabbit monoclonals against occludin (1:1000; Abclonal), claudin-1 (1:1000; Abclonal), COX-2 (1:1000; Abcam), caspase-3 (1:1000; CST), Bcl2 (1:1000; Abclonal), Bax (1:1000; Abclonal), HMGB1 (1:1000; NOVUS), Nrf2 (1:1000; NOVUS), HO-1 (1:1000; Abclonal), and mouse polyclonals against GAPDH (1:1000; Abcam) and β-actin (1:1000; Abcam) at 4 °C overnight. After incubation with the primaries, the membrane was washed three times, then incubated with goat anti-rabbit (1:5000; Zhongshan Golden Bridge, China) and goat anti-mouse (1:5000; Zhongshan Golden Bridge, China) IgG antibodies at room temperature for 1 h, and then washed three times. Lastly, the membranes incubated with enhanced chemiluminescence were exposed to the ChemiDoc Touch Imaging System. The grayscale values of protein bands were analyzed using Image J.

### 3.9. Statistical Analysis

All data were expressed as mean ± standard deviation (SD). One-way analysis of variance was used to determine significant differences between groups. Statistical analysis was performed using Statistical Program for Social Sciences (SPSS) software version 22.0 (IBM Corporation, Armonk, NY, USA). *p* < 0.05 was considered to show statistical significance.

## 4. Discussion

Patients suffering from TBI are vulnerable to gastrointestinal tract complications, which include dysphagia, gastrointestinal tract bleeding, fecal incontinence, immunosuppression, and infections. These complications affect the prognosis of TBI; therefore, alleviating intestinal secondary injury contributes to improving the prognosis of TBI [26]. The results of the HE staining experiments indicated the accumulation of leucocytes, villus atrophy, and Greenhaven’s gap in the intestinal mucosa. Chiu’s scores peaked on day 2 after TBI and then began to decline, indicating that intestinal injury was most severe at this time and that there was subsequent self-healing. The above results indicate that TBI led to significant time-dependent alterations in the intestinal mucosa. The body weight of mice is closely related to gut function. After TBI, there was transient weight loss, with the weight reaching the lowest point on day 3 after TBI and then rebounding gradually. ANXA5 effectively inhibited apoptosis levels in the intestine. This may be the mechanism through which the function of the intestinal barrier is restored, which promotes appetite and nutrient absorption. Thus ANXA5 treatment effectively promoted weight recovery.

In previous studies, the intestinal mucous membrane structure and barrier function were markedly damaged after TBI [27]. This is one of the most significant pathological changes in the intestine. Tight junction proteins, including occludin and claudin-1, are essential for maintaining villi integrity. TBI damaged claudin-1 and occludin and induced intestinal barrier dysfunction [3]. In the current study, Western blotting and immunofluorescence staining confirmed that the degradation of the tight junction-associated proteins occludin and claudin-1 was alleviated by ANXA5. Elevated MMP-9 in the intestine was also reduced by ANXA5. MMP-9 is involved in tissue destruction, and increased MMP-9 activity implies an increased disruption of barrier proteins. The above-described results indicate that the mechanism of protection of the intestinal barrier by ANXA5 may involve inhibition of MMP-9 activity.

TBI can lead to increased intestinal inflammatory factors and inflammation infiltration [3,28]. Inflammatory factors, which include the pro-inflammatory cytokines IL-1β, TNF-a, and IL-6, and the anti-inflammatory cytokine IL-10, were estimated in intestinal tissue via ELISAs. There was no significant change in the level of TNF-α on day 2 after TBI, indicating that TNF-α was not involved in intestinal damage. MPO, COX-2, and MMP-9 are markers of intestinal inflammation [29,30,31]. MPO is a functional marker and a marker of neutrophil activation, and it was investigated in the current study. It is involved in many processes that regulate inflammatory responses. The extensive release of MPO results in the formation of massive amounts of superoxide and oxides, causing tissue cell damage at the inflammatory site. Inhibiting excess MPO helps to reduce inflammatory tissue damage. In the present study, ANXA5 administration was associated with a significant reduction in MPO, indicating that intestinal inflammation was significantly improved. There was also an evidently lower COX-2 expression in the intestinal epithelium on day 2 after TBI in the ANXA5 treatment group. COX-2, another important inflammatory modulator, is present in normal tissue cells with extremely low activity. When the cells are stimulated by inflammation, COX-2 expression increases dramatically, leading to inflammation and tissue damage. In the current study, ANXA5 effectively inhibited tissue damage, evidently by reducing COX-2. HMGB1, which functions via damage-associated molecular patterns released by dying cells or activated immune cells, is a nuclear protein expressed in almost all eukaryotic cells [32]. In a previous study, ANXA5 inhibited TLR4-mediated immune responses by blocking the interaction between HMGB1 and TLR4/MD2 [20]. In another study, under intestinal ischemia-reperfusion (I/R) stress, HMGB1 released from necroptotic enterocytes drove infiltrated neutrophils to form neutrophil extracellular traps (NETs), consequently promoting inflammatory injury [33]. In the present study HMGB1 was dramatically increased in the gut after TBI, and this could be reversed by ANXA5 treatment. ANXA5 also inhibited the release of HMGB1 and contributed to the attenuation of inflammation. In summary, ANXA5 played a beneficial and anti-inflammatory role during the intestinal recovery process after TBI.

As well as an inflammatory reaction, oxidative stress occurs simultaneously in the intestine after TBI. In the current study, excessive reactive oxygen species (ROS) emerged after TBI, and acted as a cellular mediator that induced intestinal injury [34,35]. Excessive ROS interact with polyunsaturated fatty acids in biological membranes and cause lipid peroxidation. The body regulates multiple antioxidant enzymes, such as catalase and SOD, facilitating the clearance of ROS. However, the balance of the oxidant/antioxidant system could be disrupted by excessive ROS; then, lipid peroxidation and intestinal cell apoptosis emerged. In the present study, immunofluorescence staining of 8-OHDG indicated that oxidative stress levels increased significantly after TBI. ANXA5 treatment significantly reduced oxidative stress levels, suggesting that less ROS were present and that lipid peroxidation caused by ROS was reduced. MDA is an intermediate outcome of the lipid peroxidation reaction [36,37,38], and the tissue MDA content can reflect the lipid peroxidation status. In the current study, ANXA5 reduced tissue MDA levels and increased the SOD and catalase activity. This suggested that ANXA5 could alleviate peroxidative damage. We also examined intestinal tissue 4-HNE levels. 4-HNE, which is formed in the cascade process of lipid peroxide initiated by oxidation, was used as an excellent biomarker of lipid peroxidation. The contents of 4-HNE were markedly increased after TBI. In contrast with the TBI group, ANXA5 treatment significantly reduced 4-HNE expression. The activities of SOD and catalase, two indicators that may represent the capacity of the resistance to oxidation and deoxidization, were also assessed in the present study. ANXA5 enhanced SOD and catalase activity after TBI, indicating that the intestinal tract possessed a stronger antioxidant capacity. Considering all of the above-described elements, the observed decrease in lipid peroxidation may be dependent on the ability of ANXA5 to increase SOD and catalase antioxidant activity.

Nrf2 is a core transcription factor that is a component of the antioxidant response system that regulates the activity of antioxidant, cytoprotective, and detoxification genes [39]. When subjected to oxidative stress, Nrf2 can translocate to the nucleus and activate antioxidant response elements, thereby producing downstream antioxidant proteins such as HO-1 and defending against oxidative stress. The overexpression of ANXA5 may activate the Nrf2/HO-1/NQO1 antioxidant pathway and significantly attenuate Di-N-butylphthalate-induced oxidative stress [40]. In the present study, ANXA5 increased Nrf2/HO-1 expression after TBI, indicating that ANXA5 may mediate HO-1 via Nrf2 translocation and thus reduce intestinal injury. HO-1 is an anti-inflammatory and antioxidant protective enzyme. Once Nrf2 is stimulated and activated, the expression of HO-1—which is downstream of Nrf2—can significantly increase in tissues. Nrf2 can be induced by oxidative stress, ischemia-reperfusion, bacterial lipopolysaccharide, and cytokines. Many recent studies have indicated that lipopolysaccharide, radiation, burn shock, and ischemia-reperfusion could induce the anti-inflammatory and cytoprotective effects of HO-1 on intestinal injuries [41,42]. In the present study, ANXA5 treatment significantly upregulated the expression levels of Nrf2/HO-1, which has anti-inflammatory and antioxidant capacities, and reduced the severity of intestinal injury after TBI.

## 5. Conclusions

ANXA5 alleviated intestinal mucosa damage and epithelial barrier dysfunction induced by TBI. It played a protective effect by inhibiting apoptosis, inflammation, MMP-9 activity, and oxidative stress in the intestine after TBI. Moreover, ANXA5 increased anti-inflammatory factors and the activity of antioxidative enzymes. In summary, ANXA5 plays an anti-inflammatory and antioxidant role via the upregulation of the Nrf2/HO-1 pathway and downregulation of pathogenic factor HMGB1, which indicates that ANXA5 attenuates TBI-induced intestinal injury by regulating the HMGB1/Nrf2/HO-1 pathway. Thus, ANXA5 may be a promising therapeutic candidate for protecting against TBI-induced intestinal injury.

## Figures and Tables

**Figure 1 molecules-27-05755-f001:**
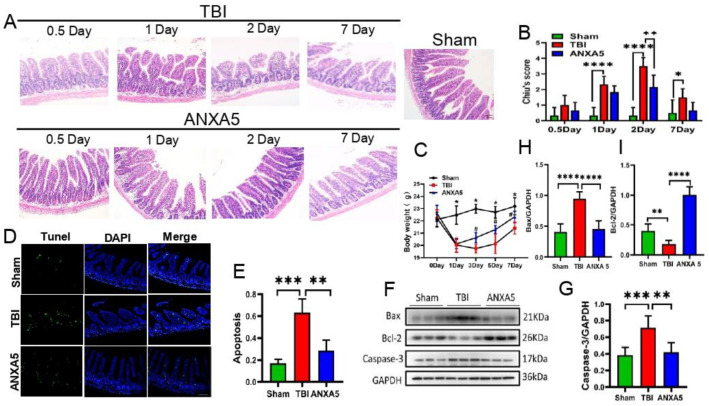
ANXA5 reduces intestinal lesions after TBI. (**A**) Representative images of ileal HE staining (×200 magnification) at 0.5, 1, 2, and 7 days after TBI (scale bar = 40 µm). (**B**) Chiu’s scores of the intestinal mucosa. Intestinal injury peaked on day 2 after TBI and then declined (*n* = 6 mice/group). (**C**) Compared with TBI group, the weight of mice in ANXA 5 treatment group increased faster, and the body weight increased significantly on day 5 post-injury. (*n* = 10 mice/group). * *p* < 0.05 vs. sham group; # *p* < 0.05 vs. TBI group (repeated measures analysis of variance followed by Bonferroni’s post hoc test). (**D**,**E**) Representative immunofluorescence images of TUNEL staining in ileum tissue (scale bar = 100 μm). The average number of TUNEL-positive cells (in green) per ×200 field was quantified. ANXA5 treatment significantly reduced apoptotic cells compared to the TBI group (*n* = 6 mice/group). (**F**–**I**) Immunoblotting showed that intestinal caspase-3 and Bax increased significantly after TBI but decreased significantly after ANXA5 treatment on day 2 after TBI. Conversely, Bcl2 decreased significantly after TBI, and ANXA5 treatment could reverse this trend (*n* = 6 mice/group). Data are expressed as mean ± SD. * *p* < 0.05, ** *p* < 0.01, *** *p* < 0.001, **** *p* < 0.0001.

**Figure 2 molecules-27-05755-f002:**
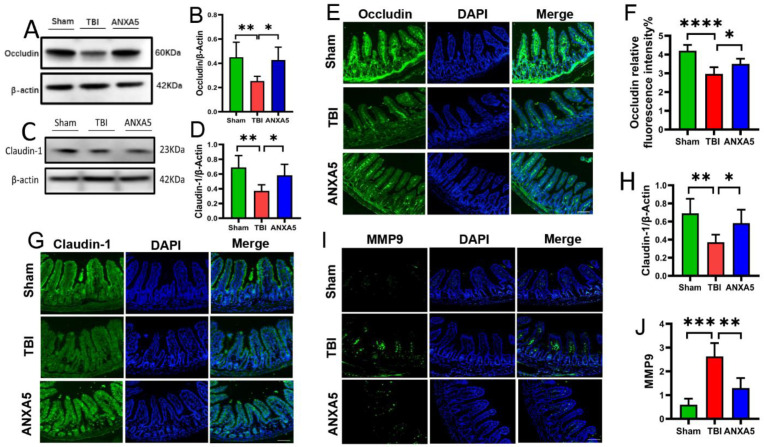
ANXA5 attenuated tight junction protein damage and inhibited MMP-9 activity in the intestine after TBI. (**A**–**D**) Western blotting showed that the expression levels of occludin and claudin-1 were significantly reduced on day 2 after TBI, but they were preserved after ANXA5 treatment (*n* = 6 mice/group). (**E**–**H**) Immunofluorescence analysis was performed to quantify occludin and claudin-1 protein expression in more detail. The mean fluorescence intensities of occludin and claudin-1 proteins (in green) in the TBI group were significantly weaker than in the sham group, and ANXA5 treatment reversed these changes (scale bar = 100 μm, *n* = 6 mice/group). (**I**,**J**) Immunofluorescence showed that ANXA5 significantly reduced MMP-9 expression in the intestine (in green) induced by TBI (*n* = mice/group). Data are shown as mean ± SD. * *p* < 0.05, ** *p* < 0.01, *** *p* < 0.001, **** *p* < 0.0001.

**Figure 3 molecules-27-05755-f003:**
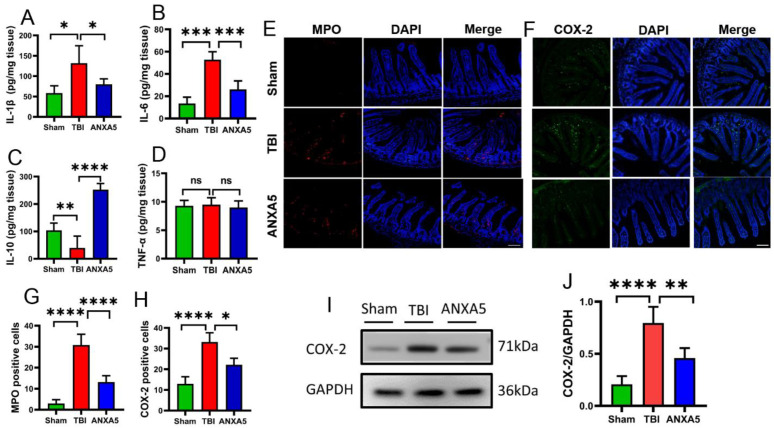
ANXA5 alleviated intestinal inflammatory response after TBI. (**A**–**D**) ELISA results showed that ANXA5 decreased the pro-inflammatory cytokines IL-1β and IL-6, but increased the anti-inflammatory cytokine IL-10 in the intestine on day 2 after TBI (*n* = 6 mice/group). (**E**–**H**) ANXA5 decreased the numbers of MPO-positive and COX-2-positive cells (in red) in the intestine on day 2 after TBI. Insets show merged images of MPO and COX-2 staining with DAPI-stained nuclei (scale bar = 100 μm, *n* = 6 mice/group). (**I**,**J**) Western blotting showed that ANXA5 treatment significantly reduced the expression of COX-2 (*n* = 6 mice/group). Data are shown as mean ± SD. * *p* < 0.05, ** *p* < 0.01, *** *p* < 0.001, **** *p* < 0.0001.

**Figure 4 molecules-27-05755-f004:**
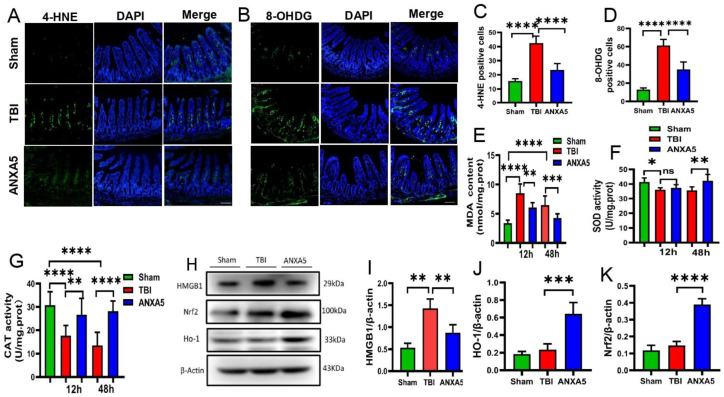
ANXA5 attenuated oxidative stress and enhanced antioxidant activity in the intestine after TBI. (**A**–**D**) Immunofluorescence showed that ANXA5 dramatically reduced the numbers of 4-HNE and 8-OHDG-positive cells (in green) in the intestine on day 2 after TBI. Insets show merged images of 4-HNE and 8-OHDG staining with DAPI-stained nuclei (scale bar = 100 μm). *n* = 6 mice/group. (**E**–**G**) ANXA5 significantly increased SOD and catalase activity and reduced the level of MDA in the intestine at 0.5 and 2.0 days after TBI (*n* = 6 mice/group). (**H**–**K**) Western blotting showed that ANXA5 significantly decreased HMGB1 expression (*n* = 6 mice/group) and increased the levels of Nrf2 (*n* = 6 mice/group) and the downstream molecule HO-1 (*n* = 6 mice/group). Data are shown as mean ± SD. * *p* < 0.05, ** *p* < 0.01, *** *p* < 0.001, **** *p* < 0.0001.

## Data Availability

All datasets generated in this study are included in the article.

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
