# Peer review of "Recombinant Human Annexin A5 Alleviated Traumatic-Brain-Injury Induced Intestinal Injury by Regulating the Nrf2/HO-1/HMGB1 Pathway"

_molecules, 2022, doi:10.3390/molecules27185755_

Round 1
Reviewer 1 Report
The authors investigated the effect of ANXA5 on TBI-induced intestinal injury and discovered that it protects against it by inhibiting oxidative stress and inflammatory responses. The mechanisms involved Nrf2/hemeoxygenase-1-induced antioxidant system activation and HMGB1 pathway suppression. It’s good work but some of the following queries need to be addressed
1. I would like to see the expression of TNF, IL1 beta, IL6, and IL10 via RT PCR
2. It's recommended to check other apoptosis markers like BCl2, and Bax via RT-PCR or blot
3. ANXA5: Show ANXA5 is not toxic to cells
4. Authors should repeat bots of Nrf2; they are not clear
5. Authors should check all spelling errors ex: the Figure 1G: check the spelling of caspase-3.
6. Please provide the ethical approval number.
Author Response
Thank you for your affirmation of our preliminary work and as for supplementary materials, please see the attachment.
Your kind considerations will be greatly appreciated.

Reviewer 2 Report
The manuscript is well-written and clear. The assays employed are appropriate and the statistical analyses are sounds. The study adds to our understanding of TBI and presents a novel way by which it might be treated. What makes the paper useful is the detail in which the functions of ANXA5 are analyzed, from basic histology, to cytokine levels, to markers of oxidative stress. The basic parameters of ANXA5 functions are well defined, making the study interesting and complete. I recommend that it be accepted.
Author Response
Thank you for your affirmation of our preliminary work.
Your kind considerations will be greatly appreciated.